



**Stratification observed by the in situ measurements**
**from the Swarm satellites**
Xiuying Wang[1], Wanli Cheng[2], Zihan Zhou[1], Dehe Yang[1], Jing Cui[1], Feng Guo[1]
1) Institute of Crustal Dynamics, China Earthquake Administration, Beijing, China
2)Xinyang Station, Henan Earthquake Administration, Henan, China
Corresponding author:   Xiuying Wang (652383915@qq.com)
Abstract: Stratification phenomenon is investigated using the simultaneous in situ plasma density
measurements obtained by the Swarm satellites orbiting at different altitudes above F2 peak. For
the first time, the continuous distribution morphology and the exact locations are obtained for the
nighttime stratification, which show that the stratification events are centered at the EIA (equatorial
ionization anomaly) trough and extend towards the two EIA crests with the most significant part
being located at the EIA trough. Another new discovery is the stratification in southern mid-latitudes;
stratification events in this region are located on a local plasma peak sandwiched by two lower
density strips covering all the longitudes. The formation mechanism of the stratification for the two
latitudinal regions is discussed, but the stratification mechanism in southern mid-latitudes remains
an unsolved problem.
Key words: stratification, ionospheric F2 layer, in situ plasma density, Swarm satellites, southern
mid-latitudes stratification, stratification morphology

## 1.   Introduction

21       Stratification is a kind of phenomenon appeared in the ionospheric F2 layer at low-latitudes near

geomagnetic equator, where additional layer is shown above the F2 layer peak due to the combined
effect of the upward E×B drift at the geomagnetic equator and the magnetic meridional neutral wind
(Balan et tal., 1997, 1998; Jenkins et al, 1997).

25       Since stratification was first reported in the mid-20th century (Sen, 1949; Skinner et al., 1954),

many studies have been conducted to study the formation mechanism, diurnal, seasonal and solar
activity dependence of this phenomenon using different measurements, such as ground-based
ionospheric sounding ionograms (Balan et al., 1997; Batista et al., 2002; Jenkins et al., 1997; Zhao
et al., 2011a), ground-based TEC (Thampi et al., 2005), satellite-based ionospheric sounding
ionograms (Depuev and Pulinets, 2001; Karpachev et al., 2013; Lockwood and Nelms, 1964),
satellite-based radio occultation (RO) observations (Zhao et al., 2011b), satellite-based in situ
measurements (Wang et al., 2019). All these studies have shown that stratification above F2 peak is
a regular rather than an anomalous phenomenon appearing both during the day and at night and is
limited in a narrow zone near the geomagnetic equator regions, and the occurrence of this
stratification phenomenon depends on season, solar activity and geomagnetic activity (Balan et al.,
2008; Batista et al., 2002; Jenkins et al., 1997; Zhao et al., 2011a).

37       The features and formation mechanism of the ionospheric F2 layer stratification have been

extensively investigated for several decades, but unsolved problems, such as the exact locations and
distribution morphology that are useful to understand this phenomenon, are still existed due to the
scattered and limited spatial coverage of the observations used in previous studies. So far, most of
these studies are based on ground-based or satellite-based ionograms. For the former, stratification





can only be observed during the period when the peak density of the stratification layer exceeds that
of the F2 layer; and for the later, only during the period when the peak density of the stratification
layer is lower than the F2 peak. Continuous global distribution of the stratification cannot be
obtained from these scattered observations though local season and solar activity dependence
features can be obtained from these long-term observations. Moreover, there are contradictory
results in these studies. Whereas, simultaneous satellite-based in situ observations at different
altitudes above the F2 layer peak can provide spatial coverage of more extensive region, which can
incorporate all local time and longitudes. And the most important is that the morphology of the
stratification along the latitudinal direction can be obtained using the continuous measurements.
In this paper, for the first time, the in situ plasma measurements from the Swarm satellites are
used to study the precise locations, distribution and morphology of the stratification phenomenon.
Nighttime stratification on the southern mid-latitudes is found, which is never mentioned in previous
studies. Our results can provide new perspective for the stratification phenomenon, which is helpful
to the insight of the ionospheric F2 layer.
**2.  Data and method**
Swarm, launched on 22 November 2013 by the European Space Agency (ESA), is a constellation
mission comprising three identical satellites (A, B & C). The three satellites are placed in two
different polar orbits, two flying side by side (A &C) at an altitude of about 460km with longitudinal
separation of about 1.4°, and a third (B) at an altitude of about 510km (Knudsen et al, 2017).
Consistent in situ plasma densities are measured by the Langmuir probes (LP) onboard the three
Swarm satellites with a time resolution of 2Hz (Lomidze et al, 2018).
The three satellites began to separate in altitudes from the end of January 2014 and the separation
operations were completed in April 2014, as shown in Fig. 1. During and immediately after the
separation operations, the three satellites orbit at similar local time and similar locations in different
altitudes above the F2 peak region, which provides a good opportunity to check and understand the
distribution and morphology of the F2 layer stratification phenomenon using simultaneous in situ
measurements obtained on a global scale.

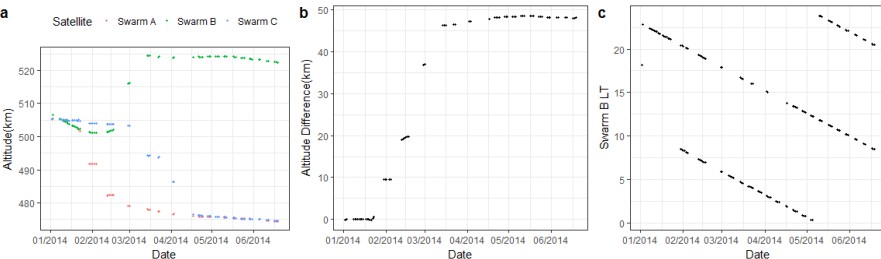


29                      Fig. 1 Variation of orbiting parameters with time

((a) Variations of orbiting altitude of the three satellites with time; (b) Variations of altitude difference between
Swarm B and A with time; (c)Variations of SwarmB LT with time. The orbiting altitude, altitude difference, and
Swarm B LT in (a), (b), and (c) indicate the daily average values near the geographical Equator.)
Separation of the three satellites follow different schemes as shown in Fig. 1(a). To simplify the
calculation, we use only the measurements from Swarm A and B as the two satellites are closer to
each other, and they have more co-located orbit tracks after altitude separation. Therefore, co-





located in situ plasma density measurements from Swarm A and B are selected using the criteria
defined below to conduct this study.
We also give the solar and geomagnetic activity indices during the select period, as shown in
Fig.2. According to Fig.2(a) and (b), the solar activity during the selected period is medium. Since
there are fewer geomagnetic events as shown in Fig.2(c), we won't distinguish the data into
disturbed and undisturbed cases here. Therefore, all the selected co-located orbit pairs are used in
this study.

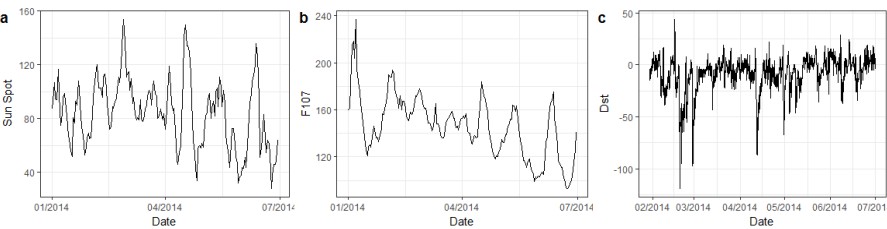

9                            Fig. 2 Variation of solar and geomagnetic indices with time

( (a) Variations of Sun spot index with time; (b)Variation of F10.7 index with time; and (c) Variation of Dst index
with time.)
To detect stratification events, measurements of the co-located orbit pairs from the two satellites
are compared directly. Spatial and temporal criteria to search co-located orbit tracks are defined as:
(1) longitude difference between two orbit tracks near the equator region is within 5°, which can
keep true for nearly all the mid-latitudes as the orbit tracks of the Swarm satellites are almost parallel
to longitude for mid- and low-latitudes. This spatial difference is reasonable considering the
longitudinal correlation can vary from 23° at mid-latitudes and 15° at low-latitudes during the day
to 11° at mid-latitudes and 10° at low latitudes during the night (Shim et al., 2008); and (2) time
difference of measurements at similar latitude between two orbits is less than 30 min as appearance
of stratification is normally much longer than this criterion (Balan et al., 1997); moreover, variations
of electron densities within 30 minutes can be neglected comparing to the diurnal variation under
geomagnetic quiet conditions.
A search of the dataset from Swarm A and B using the criteria, 1313 matched orbit pairs are found
from January to June 2014. Here, matched orbits indicate ascending (from south to north) or
descending (from north to south) half orbit tracks as a satellite passes the same location twice a day,
corresponding to daytime and nighttime respectively, as shown in Fig.1(c), which gives the local
time (LT) of Swarm B for both ascending and descending orbit during the selected data period.
Using these co-located orbit pairs, stratification events are identified only when the data
differences between Swarm B and Swarm A are positive and the positive data difference can
maintain a continual latitude of at least 5°. The morphology along the latitude, the location, and the
global distribution of the stratification events are then studied based on the detected events.
**3. Results**
The global distribution of the detected stratification events from all the co-located orbit pairs from
January to June 2014 is given in Fig. 3, also given in this figure is the variation of the occurrence
number with local time and month. As more than one event may be detected from one orbit track,
Fig. 3(a) plots all the detected stratification events, but only one event is counted per orbit track in



Fig.3(b) and (c) when comparing the statistical results. The location of each stratification event is
identified as the place where maximum data difference is located, and the color of each point
represents occurrence month of that event.

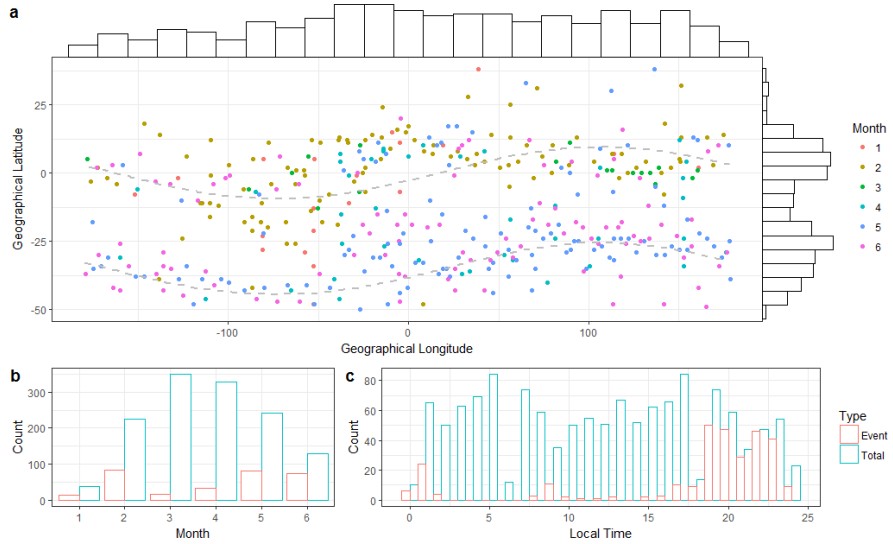

Fig. 3 Distribution of detected stratification events and variations of occurrence number with local time and month
(In (a), each point represents a stratification event, and the location of each point indicates the position of the maximum data difference
of that detected event. Color of each points indicates the occurrence month of the stratification event. Dash line indicates the geomagnetic
equator and 35° S geomagnetic latitude. In (b) and (c), 'Event' indicates detected stratification events, 'Total' indicates total co-located
orbit number.)
As shown in Fig. 3(c), most of the detected stratification events are mainly concentrated during
18:00 to 01:00 LT, with 18:00 to 23:00 LT being the most clustered period. In contrast, daytime
stratification events are fewer, which is quite different from those studies that stratifications are
concentrated from morning to noon period (Balan et al., 1998; Batista et al., 2002; Rama Rao et al.,
2005). As to the seasonal variations in Fig.3(b), the most frequent occurrence of the stratification is
in January, February, May and June when comparing to the total number of co-located orbit pairs.
However, it is necessary to point out that the local time of the two satellites coincides to dawn and
dusk during March and April, as shown in Fig.1(c), which may be the reason why detected events
are fewer during this period. Fewer stratification events during March and April is consistent with
the fewer events during the day, which we will discuss in Section 4.
It can be seen clearly from Fig.3(a) that stratification events are concentrated on two geomagnetic
latitudes, one is the geomagnetic equator region, where most previous studies are focus on; and the
other is the mid-latitude region on the Southern Hemisphere, where the distribution of the
stratification events also show the feature of being parallel to the geomagnetic equator. Geomagnetic
control of the stratification events in southern mid-latitudes is obviously shown according to its
distribution feature. In Fig.3(a), stratification events near the geomagnetic equator can occur in each
month from January to June, whereas stratification on the southern mid-latitudes only occur in May
and June, just the northern summer time.
As to the longitudinal distribution, stratification events can cover all longitudes, generally
showing a slightly denser distribution in the eastern hemisphere than in the western hemisphere,



which may be related to the limited statistical data. The all longitude coverage feature is consistent
with the results from Zhao et al. (2011b). A longitudinal peak is shown between longitude (-50,0)
in Fig.3(a), though it is not very obvious. This longitude peak coincides with the region where the
geomagnetic equator transits from the south to the north of geographic equator. This region is also
the place where the ground-based sounding observations are used by many previous studies (Balan
et al., 1998; Batista, 2001; Jenkins et al, 1997; Zhao et al., 2011a).
To demonstrate the latitudinal distribution morphology, stratification events obtained from
continual orbit tracks observed in one day and different days are given in Fig.4 and Fig.5.
Morphology of the nighttime stratification events, located near the geomagnetic equator, shows that
the stratification is centered at the equator ionization anomaly (EIA) trough, where the geomagnetic
equator is located at or near for most event cases, and extend towards the EIA crests on both
hemisphere, as shown in Fig.4 and Fig. 5(a). The occurrence of this phenomenon can be
accompanied by or without plasma depletion as shown in Fig.4(a) and Fig.5(a). Stratification can
be seen clearly from the satellite measurements even if there are plasma bubbles, whereas it cannot
be easily identified from ground-based ionograms under this disturbed situation. Latitudinal
distribution of this phenomenon indicates that most of the nighttime stratification can cover all the
regions between the two EIA crests, which is quite different from the locations of daytime
stratification, where the occurrence position is near, but not the geomagnetic equator (Balan et al.,
1998; Bastita et al., 2001; Uetomo et al., 2011).

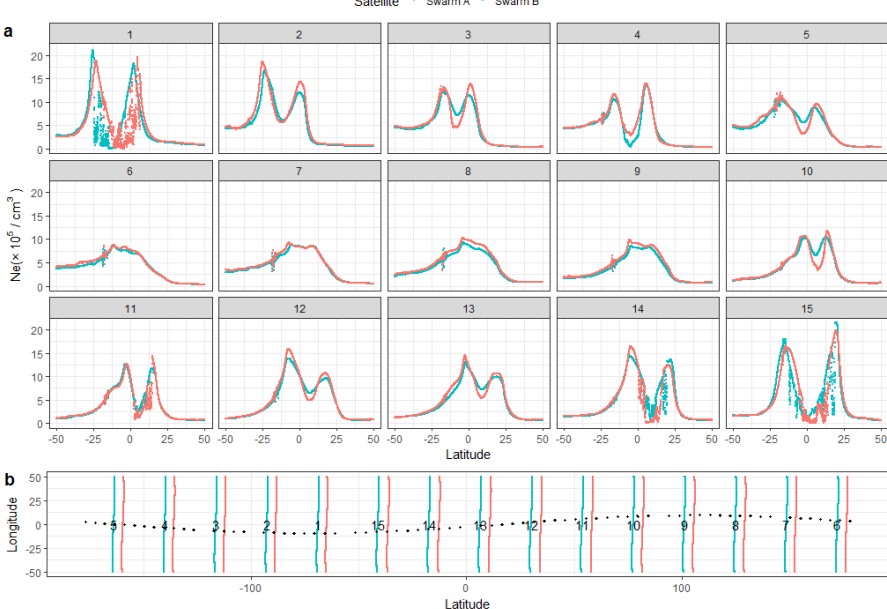


Fig. 4 Morphology of stratification examples from continual orbit tracks
((a) Morphology of nighttime stratification from continual orbits on 2014-02-03; (b) Ground tracks corresponding to the continual orbit in
(a). The dot line in (b) indicates the geomagnetic equator and numbers on the line corresponds to the numbers shown in (a).)
We also give some examples in Fig.5(b) to show the typical morphology for daytime stratification.
These latitudinal distribution morphology demonstrate clearly the results, reported by many ground-
based studies, that daytime stratification can appear on one side, Fig.5(b-1) and (b-2), or both sides,





Fig.5(b-4), of the EIA crests. We also show an example of the daytime stratification centered at the
EIA trough, which is seldom observed from ground-based ionograms. An interesting point in
daytime data is that there is a small spike centered at the EIA trough occasionally as shown in
Fig.3(d-4), which is never seen in nighttime measurements and needs further confirmation. As there
are fewer daytime stratification events, no statistical results can be obtained from these data. We
only focus on nighttime stratification in this study.

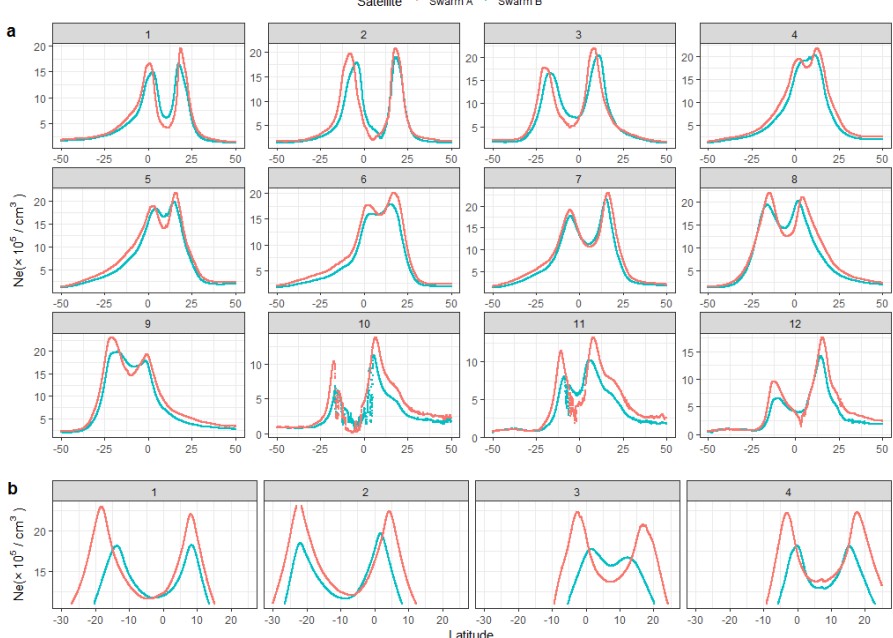

Fig. 5 Morphology of stratification examples from different LT
( (a) Morphology of nighttime stratification from different LTs. Figures 1-3 are from 2014-02-12 with LT 19:20; figures 4-9 are from 2014-
02-28 with LT 17:50; and figures 10-12 are from 2014-06-12 with LT 21:00; (b) Typical morphology of daytime stratification; Figures 1-2
is from 2014-04-22 with LT about 13:40, figure 3 is from 2014-03-22 with LT 16:00, and figure 4 is from 2014-03-20 with LT about 16:20.)
There is another southern mid-latitude regions where the detected stratification events are
concentrated on and can cover all the longitudes as shown in Fig.3(a). Stratification phenomenon in
this region is never mentioned in previous studies. All the detected stratification events in this region
occur at local nighttime in May and June as mentioned above. Typical stratification events in this
region are located on the local plasma peak along latitudinal direction, which is sandwiched by two
lower density strips as shown in Fig. 6. Stratification events in this region can occur simultaneously
with that located near the geomagnetic equator region as indicated by (11)-(12) in Fig.6. The
morphology and locations of the stratification events in southern mid-latitudes are quite different
from that in geomagnetic equator region, which may imply the different formation mechanism for
the two situations.





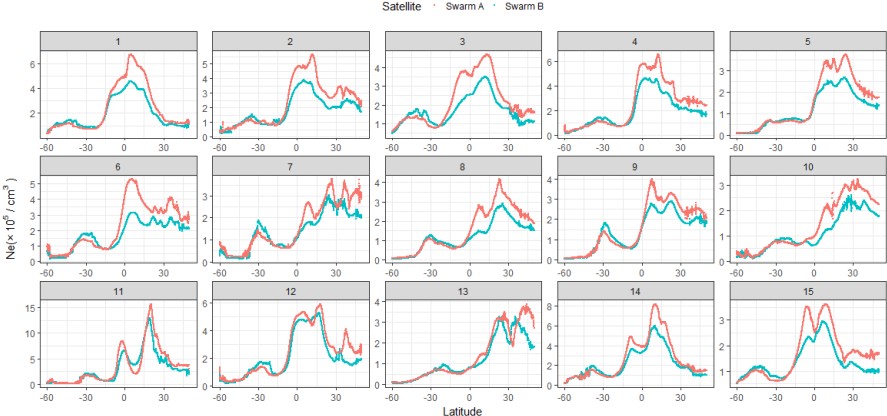

2                          Fig. 6 Examples of stratification in southern mid-latitudes

## 3    4.    Discussion

4       The continuous morphology of the stratification and its global distribution obtained in this paper

give us a more intuitive image of the stratification phenomenon and are very useful to understand
this phenomenon. But some problems requires further analysis.

7        Stratification events detected in this study are mainly concentrated during nighttime period,

which is different from previous studies that occurrence of stratification is mainly during daytime,
especially during morning to noon period when the stratification are most frequently observed from
ground-based ionograms (Balan et al., 1998; Jenskin et al., 1997; Batista et al., 2001). As to the
reason why daytime stratification events are fewer for the Swarm measurements, a possible reason
is that the altitude of the F2 peak and the stratification (F3 layer) are higher during the day than they
are at night. As suggested by Karpachev et al. (2013), the F3 layer height increases from ~450km
in the early morning to 600-750km in the afternoon, which exceeds the orbiting altitudes of the
Swarm satellites and as a result fewer daytime stratification events can be observed. This is also the
reason why there are fewer events during March and April as the orbiting LTs of the two satellites
are both during the day during this two months. In fact, nighttime stratification is suggested to be a
permanent phenomenon by studies using satellite-based iongorams (Depuev and Pulinets, 2001;
Lockwood and Nelms, 1964; Uemoto et al., 2011), the results obtained from the Swarm satellites in
this paper support this suggestion.

21        Using the measurements of the Swarm satellites, the continuous latitudinal morphology and the

exact locations of the stratification events are shown clearly for the first time, which demonstrate
that the stratification can cover all the latitudes continuously between the two EIA crests with the
most significant part being located at the EIA trough. This arch-like distribution morphology is in
accord with those studies using satellite-based observations (Depuev and Pulinets, 2001; Lockwood
and Nelms, 1964; Wang et al., 2019). Depuev and Pulinets (2001) suggest that the intensity of the
stratification has a maximum just above the equator and decreases poleward within ±10° dip; and
Lockwood and Nelms (1964) also report that the field-aligned ledge is concentrated above the
geomagnetic equator, producing a dome-like cross-section in the equatorial region. Wang et al.
(2019) also suggest an arch-like distribution of the night stratification. This feature is quite different



from the daytime stratification, which is more probable at low latitudes around the equator, rather than at the equator itself (Balan et al.,1998; Batista et al., 2003; Uemoto et al., 2011; Zhao et al., 2011b); and this location feature is demonstrated clearly by the typical daytime morphology in Fig.5(b). In contrast, the morphology of most nighttime stratification suggest that the most significant part is located at the EIA trough rather than at low latitudes on both sides. The different features of locations and distribution morphology may suggest different formation mechanism for nighttime and daytime stratification.

According to the widely accepted suggestion by Balan et al. (1998), daytime stratification (F3 layer) in equatorial region is formed due to the combined effect of the upward $E \times B$ drift and neutral wind; the upward plasma drift causes the F2 peak to drift upward and form F3 layer while the normal F2 layer develops at lower altitudes through the usual photochemical and dynamical effects. They also suggest that the upward plasma drift due to the pre-reversal enhancement (PRE) cannot form stratification because of the absence of the ionization production after sunset, which is later denied by Zhao et al. (2011a) due to the existence of post-sunset stratification. Their observations show that post-sunset stratification is different from daytime stratification due to the different solar activity and season dependence features and they suggest that post-sunset stratification is formed due to the PRE upward plasma lifts and the existence of ionization production at the high altitudes of F2 layer after sunset. However, this formation mechanism cannot explain the midnight/post-midnight stratification shown in this study and mentioned in previous studies (Depuev and Pulinets, 2001; Uemoto et al., 2011), we therefore deduce that nighttime stratification may be resulted from a different formation mechanism. According to some studies (Balan et al., 2008; Paznukhov, 2007), the mechanism responsible for the storm time stratification is similar to that in quiet periods but with a much faster processing time due to the rapid uplift of the F layer by an upward $E \times B$ drift resulting from an eastward penetration electric field. We therefore speculate that the upward plasma drift caused by the PRE will produce the same effects as that of the magnetic disturbances. As a result, plasma can be lifted up quickly by the PRE from lower altitudes to higher latitudes, which can lead to the higher densities at higher altitude and plasma depletion and plasma bubbles at lower altitude. After the PRE, the downward vertical drift resulting from the reversal electric field will replenish the depleted region by carrying the plasma from higher altitudes to lower altitude, as a result stratification can be formed during the downward carrying process at the EIA trough. At the same time, the field aligned diffusion of the uplifted plasma can maintain the EIA structure on both sides of the geomagnetic equator and form stratification at low latitudes. By this way, the nighttime stratification morphology, centering at EIA trough and extending towards the two EIA crests as shown in Fig.4(a) and Fig.5(a), can be formed. The EIA structure, which accompanies all the cases of stratification near geomagnetic equator, is supposed to be the necessary condition to form the stratification in this region. The existence of nighttime EIAs is common during geo-magnetically quiet conditions, and re-appearance of EIA is triggered by the occasionally reversed upward vertical plasma drift as nighttime vertical velocities are normally directed downwards (Yizengaw et al., 2009). In addition, the nighttime downward vertical velocities are greater after midnight than before midnight, both during magnetically quiet and perturbed times (Rajarm, 1977). Combining these results, we can explain the formation process of the nighttime stratification at and near geomagnetic equator and why most of the nighttime stratification events are concentrated between post-sunset to midnight period as shown in Fig.3(c). In addition, according to Balan et al. (2000), variations of daytime stratification arise from variations of the vertical plasma drift velocities due to the F region





zonal electric field. Similarly, the variations of the nighttime stratification may be related to the variations of the PRE amplitude, variations of the upward and downward vertical velocities, as well as variations of the frequency of the EIA re-appearance.

The new discovery in this study is the stratification on the southern mid-latitudes, which has never been mentioned in previous studies. Wang et al. (2019) propose that a small stratification may exist on southern mid-latitudes when comparing the in situ electron densities observed at different altitudes by the same payload onboard DEMETER satellite, but a definite conclusion cannot be given as the data is not observed simultaneously. The results in this paper further confirm their proposal. However, we also notice that the season and solar activity of the data used in their study are different from that in this study. Whether stratification on southern mid-latitudes can occur in all reasons or only in summer (Wang et al., 2019) or winter (in this study), both their studies and ours cannot give a definite answer due to the limit data coverage, which requires further studies when enough data are obtained. Zhao et al. (2011b) notice that there are a few cases of stratification that are far away from the geomagnetic equator in their global stratification distribution obtained from COSMIC RO data, they attribute it to the result of the propagation of atmospheric gravity waves (AGWs) often observed in mid-latitudes. We don't think their cases is similar as ours as their cases are distributed randomly on both hemisphere. As no literatures can be referenced on the stratification located in the southern mid-latitudes, a brief discussion on its possible formation mechanism is given here.

As shown in Fig.6, stratification events in this region are located on the local plasma peak, and it seems that the more obvious the local peak is, the more obvious the stratification is. Plasma enhancement in southern mid-latitudes is noticed by Tsurutani et al. (2004). They call the local peak "shoulder", and this "shoulder" can be found from TOPEX, SAC-C, and CHAMP data sets, as well as ground GPS data. Yizengaw et al. (2009) also report TEC enhancement in southern mid-latitudes and attribute it to the meridional thermospheric wind that drives the F layer plasma upward as this is the region where the wind-induced uplifting is most efficient. The morphology of the daytime "shoulder" is similar as the nighttime TEC enhancement and local peak in this study. As this local peak can exist both during the day and at night as well as under geomagnetic disturbed and quiet conditions, we suppose that it is a normal phenomenon on southern mid-latitudes. Another interesting feature from the study of Tsurutani et al. (2004) is the "shoulder" occurs only on Southern Hemisphere, similar as the feature that mid-latitudes stratification occur only on Southern Hemisphere. We speculate that the stratification in southern mid-latitudes is closely related with the local peak structure according to their common feature. Tsurutani et al. (2004) suppose the "shoulder" is likely the signature of the plasmapause, which can be used as a downward plasma source to form the stratification in the mid-latitudes, but this cannot explain why this phenomenon doesn't appear in northern mid-latitudes.

Abdu et al. (2005) suggest that precipitation of low energy (<10 keV) electrons in the SAA (South Atlantic Anomaly), namely source of ion production, together with the ionization loss process, might be a mechanism for the F2 layer stratification at mid-latitudes, but the locations of their stratification are on the southern EIA crest, quite different from the locations in this study. Moreover, precipitation mechanism cannot explain why the stratification can cover all the longitudes.

According to Lin et al. (2005), large (storm time) upward E×B drifts can lift the ionospheric layer to higher altitudes, and therefore can expand the EIA peaks to higher latitudes. However, the





proposal, transporting of equatorial plasma to higher geomagnetic latitudes by the super fountain effect, still cannot satisfactorily explain the stratification in southern mid-latitudes. For one reason, field-aligned diffusion of the uplift plasma by super fountain may lead to the mid-latitude stratification, but it cannot explain the trough between the local peak and the southern EIA crest as shown in Fig.6; the second reason, when there is no EIA signature near the geomagnetic equator, and as a result no super fountain effect, there are still many stratification cases in this region; and the third reason, this mechanism cannot explain the absence of stratification in northern mid-latitudes either. As no existing research results can satisfactorily explain the formation mechanism of the stratification in southern mid-latitudes, we put it as an open question here, and subsequent studies are anticipated.

## 5. Summary

Stratification above F2 peak is investigated in this paper using the continuous in situ plasma densities observed simultaneously by the Swarm satellites orbiting at different altitudes, some refined features and new discovery on the F2 stratification are summarized as follows:

(1) It is the first time that stratification phenomenon is investigated using direct in situ plasma density measurements.

(2) Most of the detected stratification events occur after sunset, and cluster between about 18:00 to 23:00 LT.

(3) The continuous morphology of the nighttime stratification events, located near geomagnetic equator, shows that it centers at the EIA trough and extends towards both sides, but sandwiched by the two EIA crests. This distribution feature is quite different from the daytime stratification, which located near but not the equator.

(4) A new discovery is found that some detected nighttime stratification events are concentrated near mid geomagnetic latitudes on Southern Hemisphere along all the longitudes, and the stratification in this region is found to be located on the local plasma peak. Further studies are expected on its formation mechanism.

## Acknowledgment

This work was supported by the National Key R&D Program of China (Grant no.2018YFC1503505). The plasma density data (LP) of the Swarm satellites can be downloaded from https://earth.esa.int/.

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
