# Peer review of "Stratification observed by the in situ plasma density measurements"

_Annales Geophysicae, 2019_

## Referee Comment (RC1) · Inez Batista (Referee) · 17 Feb 2020

Comments on "Stratification observed by the in situ measurements from the Swarm satellites" by Xiuying Wang by et al.

The authors use simultaneous in situ plasma density measurements obtained by the Swarm satellites orbiting at different altitudes above F2 peak to investigate the stratification phenomena at the topside F-region. The data and the assumptions are presented clearly and the manuscript has a logical presentation. Nevertheless careful language revision will contribute to a net presentation of the results. The results are interesting and deserve publication in Angeo once the comments below are attended.

1) Check the graphs in Figure 2a and 2b as a better correlation between Sunspot and

[Figure]

F10.7 should be expected.

2) Page 3, Lines 16-18: Explain the affirmation ". This spatial difference is reasonable considering the longitudinal correlation can vary from 23°atmid-latitudes and 15°at low-latitudes during the day to11°at mid-latitudes and 10°at low latitudes during the night"

3) On page 3, Lines 28-30 que authors explain the methodology to identify a possible stratification event as those that have positive gradients in plasma density between Swarm A and Swarm B maintained by at least 5 degrees in latitude. What about if the F region peak is above the satellite orbit? Is there any such cases?

4) In Figure 3a the position of the magnetic equator and of the "and 35°S geomagnetic latitude" are not correct. According to the IGRF for the year 2014 the geomagnetic equator crosses the geographic equator at approximately 196.5 degrees and 312.5 degrees, which are quite different from the positions at Figure 3a.

5) Page 4, Line 27: use "local winter" instead of "northern summer time"

6) Identify what are considered stratification events in Figures 4 and 5.

Page 5, lines 12 and 13: please refer to the frame at which the feature being described is observed.

Page 6, Line 4: there is no Fig. 3 (d4)!!!!

Page 6, Line 9-11: when referring to the distinct frames of the Figure use the word "frame" instead of "figure".

A few examples of language revision needed:

Page1, Line 21: "Stratification is a kind of phenomenon appeared..." → "Stratification is a kind of phenomenon appearing..."

Page 1, Line 40: "are still existed" → "still exists"

Page 3, Line 5: "there are fewer geomagnetic events" → "there are few geomagnetic

events"

Page 4, Line 21: "are focus on" → "are concentrated"

Page 9, Line 11: "seasons" instead of "reasons"???

Page 9, Line 16: "cases is similar" → "cases are similar"

Page 10, Line 22: "which located near but not the equator" → "which is located near but not at the equator"
* * *

---

## Referee Comment (RC2) · Anonymous Referee #2 · 12 Mar 2020

The paper presents very interesting results regarding F2 layer stratification, and also new to my knowledge. I think it should be published in this journal after some corrections and clarifications.

My main doubt is how the stratification effect is detected. I do not fully understand your explanation in page 3 (before Section 3) where you say "stratification events are identified only when the data differences between Swarm B and Swarm A are positive and the positive data difference can maintain a continual latitude of at least 5°." I think a deeper explanation is needed. In order also to fully understand Figures 4, 5 and 6, and how stratification is implied by them. May be is because I am not fully acquainted with the stratification effect, but I think that it is worth to give a more complete explanation.

Other comments:

[Figure]

I think that the F3-layer concept should be mentioned in the Introduction also. Not only in the Conclusion.

Page 1, line 23: Where it says "the magnetic meridional neutral wind", I do not think that it is a "magnetic meridional" it is just meridional. So it should be "the meridional neutral wind". Please check.

Figure 2. The Rz and F10.7 daily curves are ok, but the values of Rz I think not. Please check that the peaks reach 160 and not 200, please. I mean, I think there is a problem with the values in the y-axis of the Rz plot. Take also into account, that even though this solar cycle is a weak one, 2014 is around its maximum.

Minor corrections: Page 1, line 24: "(Balan et tal.," should be corrected to "(Balan et al.,"

Page 1, line 39: "are still existed" should be "still exist"

Page 2, line 2: "and for the later" should be "and for the latter"

Page 3, line 5: "there are fewer geomagnetic events" should be "there are few geomagnetic events"

Page 9, line 11: "can occur in all reasons or only in summer" should be "can occur in all seasons or only in summer"

---

## Author Comment (AC1) · 12 Mar 2020

Sorry for the delaying response because of some emergency event.

We thank the referee for reviewing our paper and giving us valuable suggestions.

The one by one response to the suggestion is as following.

1) F107 and Sunspot indices data were downloaded from NOAA. We re-downloaded the data, and found no mistake. We may use only one index, F107 or Sunspot number.

2) Explanation to the spatial selection criterion. As we know the topside ionosphere is variated in spatial distribution, but in a very limited spatial space, we think the variation is small enough and can be neglected. Here we select 5 degree in longitude as the

longitude criterion. To support this decision, we provide the research results that the neglected longitude variation can cover a larger extent.

We will make it clearer in the modification version.

3) We will explain the identification process in detail in the modification paper. Here a short explanation. As the plasma data from Swarm satellites is 2 Hz sampling rate, it is too complicated to compare these high sampling rate data directly. Therefore, the data along the orbit track are down sampled, and then we compare the down sampled data. For example, we calculate the average between latitude [-5,-4), and compare the average data from the two satellites. In this way, all the cases that average from Swarm B is greater than that from Swarm A are identified. To prevent the occasionally variation of the data, continuous 5 points with positive data difference, namely 5 degree latitude, is defined as an event.

As to the F2 peak height, the normal F2 layer peak height is 200 to 400km when in moderate to low solar activity condition, especially for nighttime observations. If the peak height is above the satellite height, normally stratification events occur, these are the chances to identify the events by satellite observations.

For this problem, we will give some explanation in the modification paper.

4) We are sorry for this misunderstanding. In figure 3a, the magnetic equator is from dipole magnetic coordinates. We compare it with the dipole magnetic coordinates from IGRF-2016, there is no distinct difference. We will add additional explanation to this figure.

5) We will modify the paper following this suggestion.

6) Sorry for the mistake. It is Fig. 3(b-4). We will improve the paper as suggested in the modification version.

7) We will modify the grammar error in the revision version. Sorry for poor English.

---

## Author Comment (AC2) · 12 Mar 2020

We re-check the solar activity indices data, and will give new plots in the modification paper.

---

## Author Comment (AC3) · 12 Mar 2020

We thank the referee for reviewing the paper and giving us valuable suggestions.

The one by one response to the suggestions is given below.

(1) We will give a detail description on the process to detect the stratification event in our modification paper.

A short explanation is given here.

As the plasma data from Swarm satellites is 2 Hz sampling rate, it is too complicated to compare these high sampling rate data directly. Therefore, the data along the orbit track are down sampled, and then we compare the down sampled data. For example,

we calculate the average between latitude [-5,-4), and compare the average data from the two satellites. In this way, all the cases that average from Swarm B is greater than that from Swarm A are identified. To prevent the occasionally variation of the data, continuous 5 points with positive data difference, namely 5 degree latitude, is defined as an event.

We will add the F3 layer concept in introduction section in the modification paper.

(2) Page 1, line 23. We accept the suggestion and will modify the paper.

(3) Sorry for the mistake. We re-downloaded the indices data and will re-draw the plots in the modification paper.

(4) We thank the referee for pointing out the mistakes and grammar errors. We will modify all of them and improve the paper in the revision version.

---

## Author Response (AR1)

**Response to referees**

**Comments from referees**

**Referee 1**

The authors use simultaneous in situ plasma density measurements obtained by the Swarm satellites orbiting at different altitudes above F2 peak to investigate the stratification phenomena at the topside F-region. The data and the assumptions are presented clearly and the manuscript has a logical presentation. Nevertheless careful language revision will contribute to a net presentation of the results. The results are interesting and deserve publication in Angeo once the comments below are attended.

1) Check the graphs in Figure 2a and 2b as a better correlation between Sunspot and F10.7 should be expected.

2) Page 3, Lines 16-18: Explain the affirmation ". This spatial difference is reasonable considering the longitudinal correlation can vary from 23_atmid-latitudes and 15_at lowlatitudes during the day to11_at mid-latitudes and 10_at low latitudes during the night"

3) On page 3, Lines 28-30 que authors explain the methodology to identify a possible stratification event as those that have positive gradients in plasma density between Swarm A and Swarm B maintained by at least 5 degrees in latitude. What about if the F region peak is above the satellite orbit? Is there any such cases?

4) In Figure 3a the position of the magnetic equator and of the "and 35_S geomagnetic latitude" are not correct. According to the IGRF for the year 2014 the geomagnetic equator crosses the geographic equator at approximately 196.5 degrees and 312.5 degrees, which are quite different from the positions at Figure 3a.

5) Page 4, Line 27: use "local winter" instead of "northern summer time"

6) Identify what are considered stratification events in Figures 4 and 5.

Page 5, lines 12 and 13: please refer to the frame at which the feature being described is observed.

Page 6, Line 4: there is no Fig. 3 (d4)!!!!

Page 6, Line 9-11: when referring to the distinct frames of the Figure use the word "frame" instead of "figure".

A few examples of language revision needed:

Page1, Line 21: "Stratification is a kind of phenomenon appeared: : :" ! "Stratification is a kind of phenomenon appearing: : :"

Page 1, Line 40: "are still existed" ! "still exists"

Page 3, Line 5: "there are fewer geomagnetic events" ! "there are few geomagnetic events"

Page 4, Line 21: "are focus on" ! "are concentrated"

Page 9, Line 11: "seasons" instead of "reasons"???

Page 9, Line 16: "cases is similar" ! "cases are similar"

Page 10, Line 22: "which located near but not the equator" ! "which is located near but not at the equator"

**Referee 2**

The paper presents very interesting results regarding F2 layer stratification, and also new to my knowledge. I think it should be published in this journal after some corrections and clarifications.

My main doubt is how the stratification effect is detected. I do not fully understand your explanation in page 3 (before Section 3) where you say "stratification events are identified only when the data differences between Swarm B and Swarm A are positive and the positive data difference can maintain a continual latitude of at least 5_." I think a deeper explanation is needed. In order also to fully understand Figures 4, 5 and 6, and how stratification is implied by them. May be is because I am not fully acquainted with the stratification effect, but I think that it is worth to give a more complete explanation.

Other comments:

I think that the F3-layer concept should be mentioned in the Introduction also. Not only in the Conclusion.

Page 1, line 23: Where it says "the magnetic meridional neutral wind", I do not think that it is a "magnetic meridional" it is just meridional. So it should be "the meridional neutral wind". Please check.

Figure 2. The Rz and F10.7 daily curves are ok, but the values of Rz I think not. Please check that the peaks reach 160 and not 200, please. I mean, I think there is a problem with the values in the y-axis of the Rz plot. Take also into account, that even though this solar cycle is a weak one, 2014 is around its maximum.

Minor corrections: Page 1, line 24: "(Balan et tal.," should be corrected to "(Balan et al.,"

Page 1, line 39: "are still existed" should be "still exist"

Page 2, line 2: "and for the later" should be "and for the latter"

Page 3, line 5: "there are fewer geomagnetic events" should be "there are few geomagnetic events"

Page 9, line 11: "can occur in all reasons or only in summer" should be "can occur in all seasons or only in summer"

**Authors' response**

First of all, we thank the referees for their careful reviewing of our paper, and for the constructive suggestions on the improvement of this paper.

Our responses to the referees will be given separately in the following two section.

**Response to Referee #1**

Here are the one by one response to referee #1.

1) Check the graphs in Figure 2a and 2b as a better correlation between Sunspot and F10.7

should be expected.

**Response:** We re-download the F107 and Sunspot data from NOAA, and re-draw the plots. Both the original and the new plots are given in the revision paper.

2) Page 3, Lines 16-18: Explain the affirmation ". This spatial difference is reasonable considering the longitudinal correlation can vary from 23_atmid-latitudes and 15_at lowlatitudes during the day to11_at mid-latitudes and 10_at low latitudes during the night"
**Response:** we modified this section to make it more clearly. We use this reference to support that the 5 degree longitude spatial space is reasonable.

3) On page 3, Lines 28-30 que authors explain the methodology to identify a possible stratification event as those that have positive gradients in plasma density between Swarm A and Swarm B maintained by at least 5 degrees in latitude. What about if the F region peak is above the satellite orbit? Is there any such cases?
**Response:** we add a section to the revision paper to explain the process of detecting stratification event.
As to the F region peak problem, we add a paragraph in the Discussion Section.

4) In Figure 3a the position of the magnetic equator and of the "and 35_S geomagnetic latitude" are not correct. According to the IGRF for the year 2014 the geomagnetic equator crosses the geographic equator at approximately 196.5 degrees and 312.5 degrees, which are quite different from the positions at Figure 3a.
**Response**: The geomagnetic equator and 35S latitude are from dipole coordinates, not from dip latitude. We compare the dipole geomagnetic data used in this paper with IGRF-2016, they are similar.
The dip latitude at the southern mid-latitude in this figure is not a regular curve because of the SAA. To keep the two latitude curves in consistent format, we adopt the dipole coordinates. We also add the dip equator.

5) Page 4, Line 27: use "local winter" instead of "northern summer time"
**Response**: This is corrected.

6) Identify what are considered stratification events in Figures 4 and 5.
Page 5, lines 12 and 13: please refer to the frame at which the feature being described is observed.
**Response**: Description on how to identify stratification from the given figure is added to the revision paper. And to show the feature, dip equator is added to Fig.3(a).

Page 6, Line 4: there is no Fig. 3 (d4)!!!!
**Response**: Sorry for the mistake. It is Fig.5(d-4). It is corrected according to another suggestion.

Page 6, Line 9-11: when referring to the distinct frames of the Figure use the word "frame" instead of "figure".

**Response**: We accept the suggestion, and make some modifications.

A few examples of language revision needed:
Page1, Line 21: "Stratification is a kind of phenomenon appeared: : :" ! "Stratification is a kind of phenomenon appearing: : :"
Page 1, Line 40: "are still existed" ! "still exists"
Page 3, Line 5: "there are fewer geomagnetic events" ! "there are few geomagnetic events"
Page 4, Line 21: "are focus on" ! "are concentrated"
Page 9, Line 11: "seasons" instead of "reasons"???
Page 9, Line 16: "cases is similar" ! "cases are similar"
Page 10, Line 22: "which located near but not the equator" ! "which is located near but not at the equator"
**Response**: Sorry for these mistakes. We correct all of them, and check the whole paper.

**Response to Referee 2**

Here are the one by one response to referee #2.

My main doubt is how the stratification effect is detected. I do not fully understand your explanation in page 3 (before Section 3) where you say "stratification events are identified only when the data differences between Swarm B and Swarm A are positive and the positive data difference can maintain a continual latitude of at least 5_." I think a deeper explanation is needed. In order also to fully understand Figures 4, 5 and 6, and how stratification is implied by them. May be is because I am not fully acquainted with the stratification effect, but I think that it is worth to give a more complete explanation.
**Response**: We give the process of detecting stratification event in the revision paper, and explain why 5 degree is determined.

Other comments:
I think that the F3-layer concept should be mentioned in the Introduction also. Not only in the Conclusion.
**Response**: We accept the suggestion, and add the F3 layer concept at the beginning of the paper.

Page 1, line 23: Where it says "the magnetic meridional neutral wind", I do not think that it is a "magnetic meridional" it is just meridional. So it should be "the meridional neutral wind". Please check.
**Response:** Sorry for the mistake. We delete it.

Figure 2. The Rz and F10.7 daily curves are ok, but the values of Rz I think not. Please check that the peaks reach 160 and not 200, please. I mean, I think there is a problem with the values in the y-axis of the Rz plot. Take also into account, that even though this solar cycle is a weak one, 2014 is around its maximum.
**Response:** We re-download the F107 and Sunspot data from NOAA, and re-draw the plots. Original and new plots are given in the revision paper. They have similar morphology but

different data value.

We will use the new plots.

Minor corrections: Page 1, line 24: "(Balan et tal.," should be corrected to "(Balan et al.,"

Page 1, line 39: "are still existed" should be "still exist"

Page 2, line 2: "and for the later" should be "and for the latter"

Page 3, line 5: "there are fewer geomagnetic events" should be "there are few geomagnetic events"

Page 9, line 11: "can occur in all reasons or only in summer" should be "can occur in all seasons or only in summer"

**Response:** All the above mistakes are corrected. We thank the referee for his careful work.

**Author's changes in manuscript**

Besides the modifications on the problems suggested by the two referees, we also make some other modifications. All the changes are marked with red color and underline, which is provided in a separated file.

[revised manuscript text omitted]